# Periodontal Disease, Local and Systemic Inflammation in Puerto Ricans with Type 2 Diabetes Mellitus

**DOI:** 10.3390/biomedicines11102770

**Published:** 2023-10-12

**Authors:** Oelisoa M. Andriankaja, Reuben Adatorwovor, Alpdogan Kantarci, Hatice Hasturk, Luciana Shaddox, Michael A. Levine

**Affiliations:** 1Center for Oral Health Research (COHR), University of Kentucky College of Dentistry, Lexington, KY 40536, USA; lshaddox@uky.edu; 2College of Public Health, University of Kentucky, Lexington, KY 40536, USA; radatorwovor@uky.edu; 3The Forsyth Institute, Cambridge, MA 02142, USA; akantarci@forsyth.org; 4Center for Clinical and Translational Research, The Forsyth Institute, Cambridge, MA 02142, USA; hhasturk@forsyth.org; 5Center for Bone Health, Division of Endocrinology and Diabetes, Children’s Hospital of Philadelphia, Philadelphia, PA 19104, USA; levinem@chop.edu

**Keywords:** periodontitis, cytokines, endothelium, diabetes mellitus, inflammation

## Abstract

Periodontal disease (PD) is prevalent in type 2 diabetic condition (T2DM). Objectives: We assessed the associations between serum or gingival crevicular fluid (GCF) endothelial and inflammatory mediators and chronic PD among T2DM Hispanic adults. Methods: We enrolled 248 Puerto Rican residents with T2DM aged 40–65 years. The exposures included serum inflammatory mediators (IL-1b, IL-6, IL-10, and TNF-α), endothelial adhesion molecules, RANKL levels, and the GCF content of these analytes from a subset of 158 samples. The outcomes included the percent of sites with a probing pocket depth (PPD) ≥ 4 mm and clinical attachment loss ≥ 4 mm. Adjusted logistic regression models were fit to the categorized outcomes. Results: Increased serum IL-10 (Adj. OR: 1.10, *p* = 0.04), sICAM-1 (Adj. OR: 1.01; *p* = 0.06), and elevated serum IL-1β (Adj. OR: 1.93; *p* = 0.06) were statistically significant or close to being significantly associated with a percent of sites with PPD ≥ 4 mm. An increase in GCF IL-1α (Adj. OR: 1.16; *p* < 0.01) and IL-1β (Adj: 2.40; *p* = 0.02) was associated with periodontal parameters. Conclusions: Our findings suggested that oral and systemic endothelial and inflammatory mediators are associated with periodontal clinical parameters among Hispanic adults with T2DM.

## 1. Introduction

Diabetes mellitus (DM) is a group of metabolic disorders characterized by high blood glucose levels, or hyperglycemia. Diabetes is highly prevalent worldwide [1]. A total of 37.1 million US American adults 18 years or older were diagnosed with diabetes in 2019 [2]. Type 2 diabetes mellitus (T2DM) is the most common form of this disease and accounts for 90–95% of DM. T2DM results from insulin resistance and deficiency in compensatory insulin [3,4]. T2DM produces organ and tissue damage due to its complications [5]. One of these complications is periodontal disease (PD), which is considered to be the sixth most frequent complication of diabetes [6]. According to a recent meta-analysis, individuals with T2DM have a 34% increased risk of developing PD compared to individuals without T2DM [7]. This association is even more robust among Pima Indians, a population that may be comparable to the Caribbean Taino Indian ancestors of modern Puerto Ricans [8]. PD is three to four times more prevalent in individuals with T2DM compared to normoglycemic individuals [9]. The risk and severity of PD is proportional to the loss of glycemic control [10].

The biological mechanisms explaining the relationship between T2DM and PD involve a difference in oral microbiota and an exaggerated inflammatory response to bacterial challenge [11]. T2DM and PD may share underlying common biological pathways, the exacerbation of which may lead to an occurrence or a parallel occurrence (co-occurrence) of either or both health outcomes. These chronic health conditions often occur in the same individuals, acting as co-morbid conditions [12] that may work independently or adversely affect each other [10]. Host inflammatory responses and related factors (e.g., the effect of aging or the body metabolism, which possibly impact the restoration of tissue homeostasis resolution of the inflammation) may constitute the pathways underlying the co-occurrence of these diseases [10,11].

Previous works, including ours, support the idea that local inflammatory mediators that can be measured in gingival crevicular fluid (GCF) or systemic inflammatory mediators that can be measured in serum, such as IL-1α, IL-1β, IL-6, TNF-α, and IL-10, may contribute to periodontal soft- or hard-tissue destruction in individuals with T2DM [13,14,15]. In addition, hyperglycemia can act on the mesenchymal cells, including the periodontal ligament cells, osteoblasts, and osteocytes, and the activation of these cells can dysregulate the expression of the receptor activator of nuclear factor kappa-Β ligand (RANKL) and osteoprotegerin (OPG) via the NF-kB pathway, which is coupled with a reduction in alveolar bone formation and promote periodontal bone loss [16,17,18]. Moreover, the expression of factors by vascular endothelial cells may also contribute to the disease process, such as intercellular adhesion molecule 1 (ICAM-1) and vascular cell adhesion molecule 1 (VCAM-1) [19,20,21,22].

In contrast to pro-inflammatory mediators, studies have shown that PD is associated with a reduction in cells that cause the resolution of inflammation, such as T-regs or M2 macrophages, and a reduction in anti-inflammatory mediators, such as IL-10 [23,24]. The goal of this project was to examine local and systemic cytokine levels in an under-represented Hispanic population with T2DM, and determine whether systemic and local concentrations of inflammatory mediators were related to the extent of PD. Since Hispanics present with higher levels of T2DM and less is known about this population, we examined Hispanic subjects from Puerto Rico with T2DM and measured the GCF and serum levels of pro- and anti-inflammatory analytes, the marker of bone turnover, and the periodontal status of subjects with T2DM.

## 2. Materials and Methods

### 2.1. Study Population

We analyzed data from a sample of adults 40 to 65 years old with T2DM, non-institutionalized and primarily of Hispanic origin, who participated in the cross-sectional study “Lipid-lowering agents use in Periodontitis and Diabetes Study (LLIPDS).” The study was approved by the Institutional Review Board of the University of Puerto Rico (IRB # B0930116). The study was conducted in accordance with the Helsinki Declaration of 1975, as revised in 2013.

We conducted this study between April 2017 and March 2020. Participants came from the Puerto Rico Center for Diabetes (PRCD) (50%), where the majority of T2DM individuals living in the San Juan municipality (the most populated area in PR) went for their routine health checkups; from the general population (45%), of which 7% came from the San Juan Overweight Adults Longitudinal Study (SOALS) project; and from COSSMA (5%), a private decentralized island-wide health care organization. The LLIPDS inclusion and exclusion criteria are described in detail elsewhere [25,26]. Some of the relevant criteria for this study were as follows: (1) age 40 to 65 years; (2) the presence of at least four natural teeth or no braces or orthodontic appliances; (3) no major CVD; and (4) no systemic complications, chronic inflammatory diseases, or infectious diseases within the previous 6 months. The total serum samples ranged from 192 to 248, depending on the marker under study, and were available to be used in the study. We also used a subset of 166 GCF samples to measure the GCF levels of the abovementioned analytes; a total of 1 to 13 had missing values, leaving the total GCF sample size ranging from 145 to 158 (Figure 1).

### 2.2. Periodontal Outcomes

The outcomes were assessed as described in previous LLIPDS publications [25,26]. The primary outcome was the percent of sites with PPD ≥ 4 mm [27], which was categorized into low, medium, or high tertiles. The secondary outcome was the tertile (high, medium, low) of the percent of sites with CAL ≥ 4 mm. We also recorded the plaque index performed at the six pre-selected teeth [28] and bleeding on probing (BOP) at any two of the six sites examined, one from the buccal surface and one from the lingual surface, per tooth [29]. This increases the measurement accuracy as the exact blood flow source was difficult to determine due to bleeding while probing and the blood flow from one site contaminating the other sites in the pocket [26]. All measurements were performed by trained and calibrated clinicians [30].

### 2.3. GCF and Serum Endothelial Adhesion Molecules and Inflammatory Markers

All participants fasted at least 10 h before the morning blood sample draw following a standard protocol. The blood samples were centrifuged at 3000 rpm for 15 min, after which the serum was stored at −80 °C until analyzed. A total of 248 individuals had measures of serum IL-1α, IL-1β, IL-10, TNF-α, sICAM-1, sVCAM-1, and RANKL analyzed by multiplex Luminex^®^200^TM^ immunoassay or ELISA at the Multiplexing Core of the Forsyth Institute (Cambridge, MA, USA). The following were the intra-assay coefficient of variations (CVs) and the minimum sensitivity for each analyte, respectively: IL-1β: CV: 5.85%, minimum sensitivity: 0.8 pg/mL; IL-6: 5.34%, 1.7 pg/mL; IL-10: 6.79%, 1.6 pg/mL; RANKL: 13.87%, 4.7 pg/mL; TNF-α: 5.25%, 1.2 pg/mL; sICAM-1: 4.33%, 87.9 pg/mL; and sVCAM-1: 4.18%, 238 pg/mL.

We obtained a maximum of four GCF samples at the mesio-buccal inter-proximal site of the first molar of each participant quadrant. The corresponding site at the second molar, second pre-molar (PM), or first PM, in that order, was used in case the first molar was absent. The site was isolated with cotton rolls and air-dried. A PerioPaper strip (OraFlow Inc., Hewlett, NY, USA) was gently placed until it was held by the gingival margin without attempting to push it down into the pocket. The strip was placed into the crevice for 30 s before removing it. Periopaper strips contaminated with blood were discarded. Each sample was stored at −80 °C until further processing. A subset of 200 GCF samples were processed and analyzed for the levels of IL-1α, IL-1β, IL-10, TNF-α, RANKL, OPG, sICAM-1, and s-VCAM-1 using a multiplex Meso Scale Discovery immunoassay [31] or ELISA in the laboratory of the Center for Clinical and Translation Science (CCTS) at the University of Kentucky. Of the 200 GCF samples, 68 were from 34 participants with one diseased site (PPD ≥ 4 mm and CAL ≥ 3 mm) and one non-diseased site. Thus, we took the average levels from those two sites for the GCF levels of these participants. The remaining 132 GCF samples came from participants who had only diseased or non-diseased sites. The following are the intra-assay coefficient of variations (CVs, %), inter-assay CVs (%), and the minimum detectable values of each analyte, respectively: IL-1α: 6.59%, 10.27%, and 0.12 pg/mL; IL-1β: 3.43%, 6.67%, and 0.04 pg/mL; IL-10: 10.22%, 7.9%, and 0.02 pg/mL; RANKL: 10.91%, 6.53%, and 0.41 pg/mL; OPG: 7.72%, <15%, and 4.81 pg/mL; TNF-α: 9.37%, 7.87%, and 0.06 pg/mL; sICAM-1: 5.03%, 11.37%, and 1.21 pg/mL; and sVCAM-1: 5.82%, 6.47%, and 5.78 pg/mL.

### 2.4. Other Available Data

In the interest of brevity, we refer the reader to the previous LLIPDS publications for a complete list of items we collected [25,26]. A few of the more relevant items were as follows: demographic data; lifestyle health habits, including smoking status (never, former, or current), alcohol consumption (abstainer, former, or current), and exercise information (yes/no); diabetes duration (years); and detailed information on current medication use. The averages of three anthropometric measurements of waist circumference, height, and weight were obtained to the nearest 0.1 cm or 0.1 kg. Body mass index (BMI) was computed using the weight measure (in kg) and the square of the height measure (m). An average of three blood pressures was obtained. The fasting serum glucose, insulin, and lipid panel measurements were available [32], as was glycosylated hemoglobin (HbA1c %).

### 2.5. Statistical Analysis

We initially performed an exploratory data analysis to assess the pattern of distribution and the presence of outliers, and to check potential relationships between the variables of interest with the help of summary statistics and plots of the data. We described the general characteristics of and the distribution for both the GCF and the serum levels of the analytes by aspects of our primary outcome, such as the low vs. medium/high tertile categories of PPD ≥ 4 mm using the mean (standard deviation), the median (25th, 75th percentiles), or the frequency (percent). We assessed the potential correlations between the covariates and the independent variables to avoid including two highly correlated variables in the same statistical model; as per the results of this analysis, we entered the values one by one into our statistical models because most were highly correlated. We used generalized linear models with the binary family and logistic regression function to determine if there was an association between an increase in the level of each analyte and a higher level of the primary and secondary outcomes. We then added the study of the association with a binary variable among those markers, the increase in the levels of which were not associated with the outcomes, indicating whether participants’ levels of those markers were in the highest category. The analyte level was defined as the highest tertile distribution. We considered major non-dental factors for PD as potential confounding factors, including age; gender; years of education; BMI; smoking status; alcohol consumption; and exercise; as well as (for the change in estimate models) lipid panel levels, insulin, fasting glucose, or HbA1c; diabetes duration; and any self-reported use of anti-inflammatory agents, lipid-lowering agents, metformin or other glucose-lowering medications, medications intended to control insulin level, and blood pressure medications. We also added the plaque index to our models for further adjustment.

## 3. Results

The periodontal outcome parameter distributions, as well as the characteristics of the population by the primary periodontal outcome, are described in Table 1a,b, respectively. Approximately 63% of participants had a medium/high tertile percent of sites with PPD ≥ 4 mm (Table 1a).

This percentage varied between 1% and 68%, with an average of approximately 9.5%. The mean percent of sites with PPD ≥ 4 mm in the lowest tertile was 0. About 66% of the participants had a medium/high tertile percent of sites with CAL ≥ 4 mm; this percentage ranged from 1.4% to 82% with a mean of 14.7%. The mean percent of sites with the secondary outcome CAL ≥ 4 mm in the lowest tertile was 0.4%, ranging from 0 to 1.4%.

Participants with a medium/high percent of sites with PPD ≥ 4 mm were more likely to be male and less educated, and demonstrated poorer glycemic control (i.e., elevated glycated hemoglobin, HbA1c) than their participant counterparts (Table 1b).

Participants with a medium/high percent of sites with PPD ≥ 4 mm, compared to their peers with a low percent of such sites, also demonstrated poorer oral hygiene, as indicated by their higher mean plaque index and more tooth surfaces with BOP.

The distribution of the endothelial and inflammatory mediators and the inflammatory bone-related factors assessed from either serum or GCF by periodontal status appear in Table 2 and Table 3, respectively. Overall, the serum analyte levels appeared to be more prominent in those participants with medium/high levels of the percentage of sites with PPD ≥ 4 mm, compared with participants with the lowest percentage level (Table 2).

Notably, the serum levels of IL-10 (*p* = 0.04) and sICAM-1 (*p* = 0.09) were higher in medium/high percent of sites with PPD ≥ 4 mm. Regarding the GCF analytes (Table 3), the levels of most of the analytes, except sVCAM-1 and RANKLG/OPG were significantly higher in the medium/high percentage of sites with PPD ≥ 4 mm compared to that of the lowest percentage of sites with PPD ≥ 4 mm counterpart (all *p*-values < 0.05).

The crude odds ratio (OR) for the association between increase in serum IL-10 and a higher percentage of sites with PPD ≥ 4 mm was 1.09 (*p =* 0.03) (Table 4).

After we adjusted our models for age, gender, education, smoking status, alcohol status, BMI, mean plaque index, lipid-lowering agent (LLA) use, and HbA1c, the increase in serum IL-10 was associated with a 10% increase in the odds of a higher percent of sites with PPD ≥ 4 mm (adjusted (Adj.) OR: 1.10; *p* = 0.04). The crude and adjusted estimates for the association between an increase in serum sICAM-1 and a higher percentage of sites with PPD ≥ 4 mm remained borderline significant (Adj. OR: 1.00; *p* = 0.06). We did not find any statistically significant relationships between a possible increase in the serum levels of sVCAM-1, IL-1β, IL-6, TNF-α, or RANKL and the periodontal primary outcome among individuals with T2DM.

Our crude estimate for the association between an increase in serum IL-10 (mg/dL) and the periodontal secondary outcome, i.e., the percent of sites with CAL ≥ 4 mm was OR: 1.08 (*p* = 0.04). After model adjustment, this estimate lost its statistical significance (Adj. OR: 1.06; *p* = 0.15). We also did not find statistically significant associations between a possible increase in serum levels of sICAM-1, sVCAM-1, IL-1b, IL-6, TNF-α, or RANKL with a higher percentage of sites with CAL ≥ 4 mm.

The estimates for the association between the increase in GCF analyte levels and the periodontal parameters are presented in Table 5.

The crude ORs for the associations between an increase in GCF IL-1α or GCF IL-1β with a greater percentage of sites with PPD ≥ 4 mm were OR 1.19 (*p* = 0.001) and 2.91 (*p* = 0.002), respectively. This association remained statistically significant after model adjustment (Adj. OR_IL-1α_: 1.16; *p* <0.01; Adj. OR_IL-1β_: 2.40; *p* = 0.02). However, each unit increase in RANKL/OPG was borderline statistically significant and associated with a decreased adjusted OR, indicating a “protective effect” for having a greater percentage of sites with PPD ≥ 4 mm (Adj. OR _RANKL/OPG_: 0.20; *p* = 0.06). We observed no statistically significant association between each one standard deviation (SD) unit increase in GCF IL-10 or each one unit increase in TNF-α, sICAM-1, or sVCAM-1 and a larger percentage of sites with PPD ≥ 4 mm.

Similarly, an increase in GCF IL-1α or GCF IL-1β was associated with an increase in the odds of having a greater percent of sites with CAL ≥ 4 mm (Adj. OR _IL-1α_: 1.12; *p* = 0.02; Adj. OR _IL-1β_: 2.09; *p* = 0.03), while an increase in RANKL/OPG was associated with a statistically significant decrease in the odds of having a greater value of the secondary outcome (Adj. OR: 0.19; *p* = 0.04). The association between sVCAM-1 and the secondary outcome was borderline (Adj. OR: 0.99; *p* = 0.06).

We further analyzed the associations between elevated serum analyte levels or elevated GCF levels, which we defined as the highest tertile of the distribution (vs. low/medium tertiles), and the periodontal outcomes. There were borderline associations between elevated serum IL-1β (adj. OR: 1.93; *p* = 0.06) or sICAM-1 (adj. OR: 1.72; *p* = 0.08) and higher odds of having a higher percentage of sites with PPD ≥ 4 mm (Figure 2a).

The data consistently showed borderline or statistically significant associations between elevated levels of GCF IL-1α or GCF IL-1β and higher odds of having the primary outcome (Adj. OR _IL-1α_: 3.98; *p*< 0.01;Adj. OR _IL-1β_: 2.23; *p =* 0.09) or the secondary outcome (Adj. OR _IL-1α_: 3.63 *p*< 0.01;Adj. OR_IL-1β_: 2.92; *p =* 0.02) (Figure 2b above).

In addition, an elevated GCF sICAM-1 was associated with higher odds of having the PPD primary outcome (Adj. OR: 2.63; *p* = 0.04), while an elevated GCF IL-10 was associated with higher odds of having a greater value of the CAL secondary outcome (Adj. OR: 3.17; *p* = 0.01). Moreover, an elevated GCF sVCAM-1 was borderline statistically significant and associated with lower odds of having a greater percentage of CAL secondary outcome (Adj. OR: 0.39; *p* = 0.06). We did not find any association between elevated GCF TNF-α or GCF RANKL/OPG and the outcomes.

## 4. Discussion

In this study, we examined local and systemic pro- and anti-inflammatory mediator levels, markers of bone turnover and clinical periodontal parameters in Hispanics with T2DM. Our findings showed an increase in serum IL-10 and sICAM-1 levels and an elevated serum IL-1β level to be associated with a higher percentage of sites with PPD ≥ 4 mm, and that the level of association was significant or close to being significant. For GCF analyte levels, an increase in GCF IL-1α or GCF IL-1β was associated with a higher percentage of both periodontal parameters. Elevated GCF sICAM-1 was associated with only a higher percentage of PPD, while elevated GCF IL-10 was associated with only a higher percentage of CAL. Unexpectedly, an increase in GCF RANKL/OPG was associated with a lower percentage of both periodontal parameters, and an increase in level or elevated level of GCF sVCAM-1 was associated with a lower percentof sites with CAL ≥ 4 mm.

Our findings on the increase in serum IL-10 and a higher occurrence of periodontal parameters in individuals with T2DM are consistent with previous work [15]. A small clinical trial conducted to assess the effects of non-surgical periodontal therapy (NSPT) on serum IL-10 in individuals with T2DM showed a greater baseline level of IL-10, which may affect their findings [33]. In contrast, another study found no difference between baseline serum IL-10 levels across normoglycemic individuals, individuals with generalized PD, individuals who were pre-diabetic with generalized PD, and T2DM individuals with generalized PD [34]. Likewise, other studies showed no difference in serum IL-10 levels between individuals with chronic PD and T2DM compared to healthy individuals without PD or the other groups (PD+ T2DM− and PD-T2DM+) [35,36].

The elevated plasma/serum IL-10 we observed in individuals with T2DM is observed more frequently in the hyper-inflammatory states, as confirmed by a recent systematic meta-analysis [37]. Similar to COVID-19 infection, this phenomenon could be due to either a failure by IL-10 to suppress inflammation or IL-10 may have a dual role in inflammatory response in different settings [38].

We observed a trend in elevated GCF IL-10 with a higher percentage of sites with periodontal parameters, especially CAL ≥ 4 mm, indicating its possible pro-inflammatory property in T2DM+ individuals. Other studies showed a higher baseline GCF IL-10 among individuals with PD and T2DM+ compared to the other groups [34], or an elevated GCF IL-10 activity among T2DM+ individuals with severe PD compared to T2DM− individuals with the same PD status [39]. In contrast, findings in other studies showed less GCF IL-10 activity among adults with T2DM and PD compared to T2DM+ adults lacking PD, and the lowest IL-10 activity level was observed among adults with PD only [40].

Our data also showed that the increase in levels or the highest tertile of the serum and GCF levels of sICAM-1 is associated with a higher occurrence of percent of sites with PPD ≥ 4 mm. Although we did not find any association with serum sVCAM-1, we did observe a negative association between an increase in or elevated level of GCF sVCAM-1 and a low occurrence of the percent of sites with CAL ≥ 4 mm. Information concerning the association between these adhesion molecules and PD development in T2DM individuals is scarce. In another study, the group of individuals with PD and uncontrolled T2DM had the highest serum baseline sICAM-1 level, after whom came the group of individuals with PD and controlled T2DM compared to those of systemically healthy individuals with chronic PD [41]. We found no comparable report of GCF sICAM-1 or serum and GCF sVCAM-1, together with PD development in individuals with T2DM. Nonetheless, our recent findings for overweight or obese individuals showed a direct association between an elevated baseline serum sICAM-1 or sVCAM-1 and an indirect association between elevated baseline hs-CRP (via ICAM-1) and an increased risk of periodontitis at the 3-year follow-up visit [26]. In addition, elevated levels of sICAM-1 in both serum and GCF were reported in non-smokers with PD, and elevated serum sICAM-1 but not GCF sICAM-1 was observed in smokers with PD, suggesting the potential effect of smoking on the local level of sICAM-1 [42].

We did not find any associations between serum IL-1β, IL-6, or TNF-α and any of the periodontal parameters. Serum/plasma IL-β, IL-6, and TNF-α have been suggested to be elevated in individuals with both PD/gingivitis and T2DM compared to the PD+ T2DM−, PD− T2DM+, or PD− T2DM− groups [13,36,43,44]. Other researchers did not report the same trends [34,45,46].

We demonstrated that an increase in GCF IL-1α and IL-1β was associated with a large occurrence of periodontal parameters, but we did not find an association between GCF TNF-α and any of the periodontal parameters with T2DM+. In the literature, examinations concerning IL-1α and PD were mostly related to genotyping [47,48,49]. There is only a limited number of studies with a study design comparable to ours that also assessed the difference between PD+ and PD-T2DM individuals for GCF analyte levels. Perhaps the most abundant data is available for GCF IL-1β and TNF-α levels for T2DM+ and PD, but the findings remain heterogeneous, especially for GCF TNF-α. A recent systematic review and meta-analysis suggested GCF IL-1β is mostly higher in T2DM+ individuals with PD compared those with T2DM- and PD [50]. Some recent reports, including ours from the ARIC dental study data, suggested elevated GCF or baseline GCF levels of IL-1β or TNF-α in T2DM individuals with chronic periodontitis/gingivitis [13,34,51,52,53,54,55,56]. By contrast, no lower GCF TNF-α levels have been reported in T2DM individuals with chronic PD [40,50,54].

We did not find any association between the elevated serum level of RANKL and the occurrence of periodontal parameters among T2DM individuals. No difference was observed in the serum level of RANKL across individuals with PD and T2DM+ who were smokers or non-smokers and in systemically healthy individuals with PD [57]. In contrast, a higher baseline serum of RANK and RANKL/OPG and lower baseline serum of OPG were observed among individuals with PD and T2DM+ compared to those with T2DM- and PD+/− [58]. Our data showed an unusual increase in GCF RANKL/OPG associated with a lower occurrence in CAL secondary outcome. The GCF levels of sRANKL and RANKL/OPG have been observed to be up-regulated in T2DM individuals with chronic PD, and poor glycemic control appears to increase the ratio [54,59]. In summary, our findings suggested the involvement of GCF and possibly elevated serum endothelial and inflammatory mediators in a more prominent occurrence of periodontal parameters among Hispanic T2DM individuals. However, the other mediators we tested, including GCF RANKL/OPG and sVCAM-1, did not yield similar associations. Our recent results suggested an association between LLA use, a lower occurrence of periodontal parameters along with an equivalently lower serum/GCF sVCAM-1 or GCF sICAM-1, and unusually higher GCF RANKL/OPG [60]. However, as these LLIPDS participants reported taking an average of three types of diabetes-controlling medications, the medications might have influenced their analytes, which likely affected our results in unpredictable ways because little is known about such interactions. Nonetheless, our findings still suggest the presence of an inflammatory immune response imbalance against periodontal dysbiosis in individuals with T2DM.

Other than the race of the study sample, discrepancies between findings across the cited studies and ours might have been related to the differences in sample size, the study design, study populations, heterogeneity in periodontitis case definition(s) or periodontal parameter(s) employed, potential biases, and uncontrolled confounding factors. Most of the reports from the cited studies mentioned a small sample size, even if the report in question was based on the pre-PD treatment baseline data of a clinical trial on individuals with T2DM, thus limiting the interpretation of their findings. The other reasons for negative results could be the presence of unknown uncontrolled confounding factors due to the combined effects of multiple medications on the levels of the inflammatory mediators, which are difficult to disentangle.

The data in this study were collected before the most recent and inclusive classification of periodontal and peri-implant diseases in 2017 [61]. While the prevalence of periodontal disease can differ across the varieties of periodontitis and gingivitis case definitions, such as the CDC/AAP [62] or the 2017 AAP/EFP periodontitis classifications, the use of the percentage of diseased sites based on the periodontal pocket depth and clinical attachment levels instead intend to express (or determine) the extent of existing periodontal inflammation, such as the focus of our study. In addition, these parameters are typically used for each individual treatment plan and for mechanistic studies.

This is the first study to assess the effects of inflammation on periodontal health in Hispanics with T2DM. Our data provided both high-quality serum and GCF analyte measures as well as valid and reliable data with comprehensive periodontal measurements following the NHANES oral health protocol [63]. We also obtained data concerning an extensive array of factors likely related to periodontal health, and this data can be employed in further hypothesis generation. Our results may also provide insights into public health issues that need addressing among Hispanics with T2DM.

There are limitations that should be considered: Since this study was a non-probability convenience sample, our findings may not be generalized beyond the island-wide diabetic population residing in Puerto Rico. However, our examination of the main characteristics of the study population by the source of the participants (i.e., 50% T2DM participants from PRCD, 45% from the general population, and 5% from COSSMA), including age, gender, education, and smoking status, did not show any significant differences.

We examined various variables related to diabetes treatment, including medications, such as metformin and injectable insulin, as well as diabetes duration. These variables were correlated with ”HbA1c” and explained fewer outcomes than HbA1c. Therefore, we utilized the ”HbA1c” variable due to having fewer missing values compared to the other covariates. Furthermore, when we considered each covariate for a potential adjustment in our statistical models, “HbA1c” had the most significant impact on altering the associations when compared to other covariates. Due to statistical power constraints, subcategory analyses are discouraged as it might introduce a new bias and consequently be reported as a limitation.

We are aware that we might have had some selection bias affecting our results due to excluding participants with missing teeth if the reason(s) for this could be related to inflammatory mediators, their periodontal status, or both. Participants with surviving teeth may appear healthier with less PD than those with teeth extracted due to PD. However, periodontal research frequently suffers from this intrinsic bias, especially in retrospective analyses.

Recall bias for data obtained from participants via interview-based questionnaire concerning confounding factors, e.g., information on medication use, dosage, and duration of use, might also have occurred. However, we employed thorough but straightforward and neat questions on the topic of interest, intended to boost participants’ ability to remember without biasing their remembrance of the information-relevant memory due to their knowing precisely why we were asking.

Nonetheless, our use of a cross-sectional study design does not permit us to draw causal interpretations from our results due to the temporal sequence issue. Participants might have had high or low levels of periodontal parameters prior to the GCF or serum levels of the analytes we observed, yet due to our “keyhole” perception of the obtained data, we cannot work out the actual temporal sequence.

Our findings still have a substantial public health impact and value to other researchers in this area. Prevention or control of T2DM, along with improved oral hygiene control to prevent or manage oral and systemic inflammation, may also prevent or permit the management of extent PD, as these two conditions are likely related via systemic inflammation, albeit partly. The oral cavity mirrors an individual’s general health; thus, PD in the oral cavity may reflect existing, previously undetected, or exacerbated chronic systemic (or at least oral) inflammation in individuals with T2DM. On the other hand, although we did not examine this possibility, the prevention or control of PD may also help improve systemic inflammation in T2DM individuals.

## 5. Conclusions

In summary, our results suggest associations between oral and systemic endothelial along with inflammatory mediators and the more frequent occurrence of periodontal parameters in Hispanic adults with T2DM. However, further extensive prospective work is required to validate our findings.

## Figures and Tables

**Figure 1 biomedicines-11-02770-f001:**
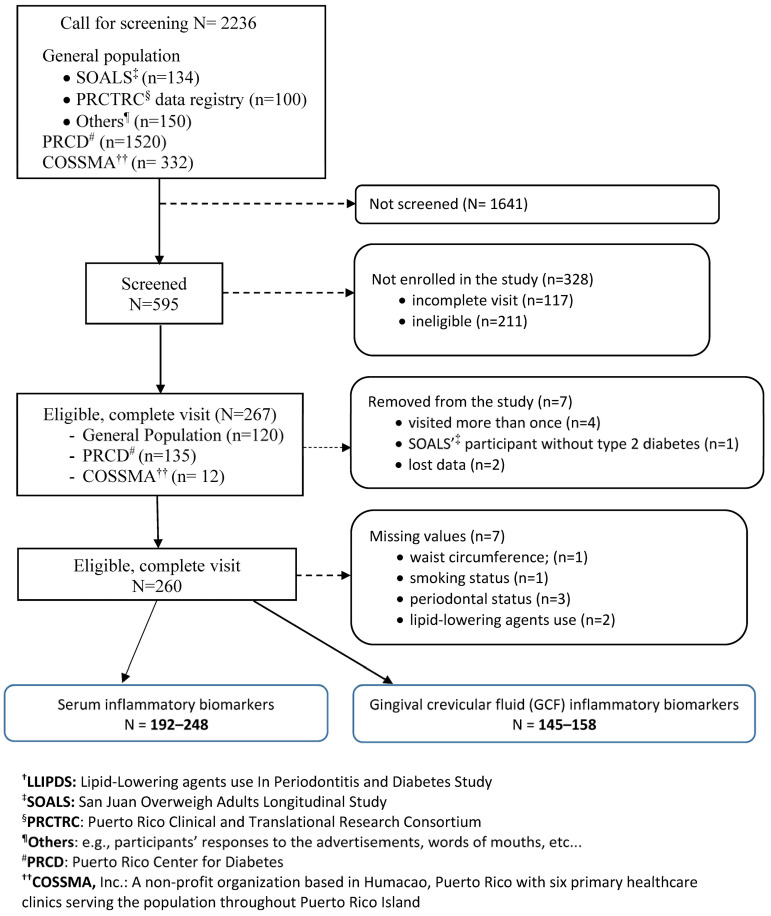
Selection of the 248 study participants, LLIPDS ^†^ (26 April 2017–9 March 2020).

**Figure 2 biomedicines-11-02770-f002:**
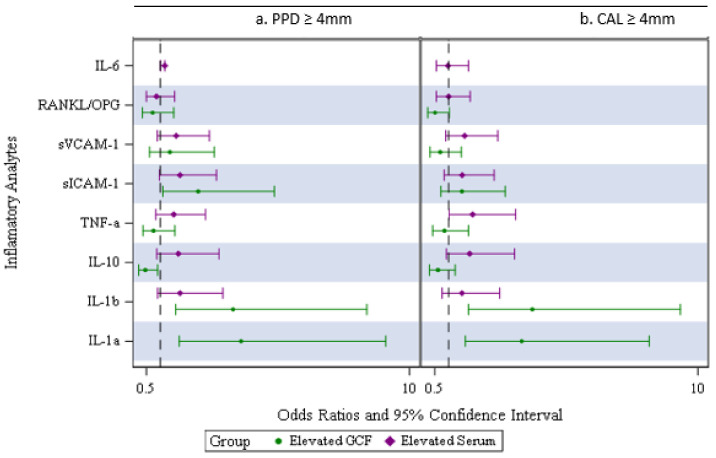
(**a,b**) Inflammatory analytes comparing the odds ratios of low/medium vs. highest percent of sites with PPD ≥ 4 mm or CAL ≥ 4 mm for serum and GCF.

**Table 1 biomedicines-11-02770-t001:** (**a**) Distribution of periodontal parameters, N = 248. (**b**) General characteristics of Hispanic adults with type 2 diabetes by periodontal status, N = 248.

(**a**)			
**Outcome**	**Tertile** **n (%)** **Mean ± SD (range) %**		
Percent of sites with PPD ≥ 4 mm	**Low**	**Medium/High**	
	93 (37%) 0%	155 (63%) 9.5 ± 12.0 (0.9–67.8)%	
Percent of sites with CAL ≥ 4 mm	84 (34%) 0.4 ± 0.5 (0–1.4)%	164 (66%) 14.7 ± 16.0 (1.41–82.1)%	
(**b**)			
**Characteristic**	**Low Tertile PD** **N = 93 (37%)**	**Medium/High Tertile PD** **N = 155 (63%)**	
	**Mean ± SD, Median** **(q1, q3), N (%)**	**Mean ± SD, Median** **(q1, q3), N (%)**	** *p* ** **-Value**
Age (yr.)	54.6 ± 5.8	54.3 ± 6.2	0.721
Male gender	30 (32.3)	78 (50.3)	**0.005**
Education (≤12 yrs.)	24 (25.8)	64 (41.3)	**0.014**
Smoking status			0.114
Never	67 (72.0)	93 (60.0)	
Former	16 (17.2)	44 (28.4)	
Current	10 (10.8)	18 (11.6)	
Alcohol drinking (Current) ^†^	37 (39.8)	68 (44.7)	0.447
Hypertension (yes)	64 (68.8)	101 (65.2)	0.555
BMI (kg/m^2^)	35.2 ± 12.1	35.1 ± 9.9	0.94
HbA1c (%)	7.7 ± 1.8	8.3 ± 2.0	**0.024**
HdL-C (mg/dL) ^‡^	49.2 ± 13.3	46.5 ± 13.0	0.108
Diabetes duration (yrs.) ^†^	9.6 ± 6.5	11.4 ± 8.5	**0.077**
LLA current users (yes) ^§^	52 (55.9)	79 (51.0)	0.45
LLA duration of use (yrs.) ^§^	0.5 (0, 3.7)	0.05 (0,5)	0.374
Anti-inflammatory agents (yes) ^†^	9 (9.7)	8 (5.2)	0.173
Mean Plaque index ^†^	0.9 ± 0.5	1.1 ± 0.6	**0.025**
Tooth brushing (≥ twice/day)	17(17.9)	27 (17.7)	0.9
Missing teeth	2.8 ± 3.5	2.2 ± 2.4	0.133
Bleeding on probing (%)	18.5 ± 10.1	25.0 ± 12.3	**<0.001**
Exercise (yes) ^†^	29 (31.5)	46 (29.9)	0.785

^†^ Missing values: diabetes duration 11, alcohol consumption 2; plaque index 7, exercise 2; ^‡^ High-density lipoprotein-cholesterol; ^§^ LLA: lipid-lowering agents. *p*-values in bolds are statistically significant (*p* < 0.05) or close to being significant (*p* < 0.10).

**Table 2 biomedicines-11-02770-t002:** Distribution of the serum biomarkers by periodontal status, N = 248.

Biomarker (pg/mL)	Low Tertile PD n = 93 (37%)	Medium/High Tertile PD n = 155 (63%)		
	Median (q1, q3)	Median (q1, q3)	*p*-Value ^‡^	Missing Value
**IL-1b**	2.8 (0.9, 4.0)	3.3 (2.3, 5.1)	0.109	56
**IL-6**	5.3 (3.8, 6.4)	5.3 (3.7, 7.2)	0.822	
**TNF-** **α**	9.6 (7.1, 12.1)	10.0 (7.1, 13.0)	0.540	
**IL-10**	8.8 (6.4, 11.5)	9.8 (7.4, 12.7)	**0.043**	45
**sICAM-1 ^†^**	44.5 (32.1, 54.5)	46.5 (36.1,72.4)	**0.093**	
**sVCAM-1 ^†^**	69.5 (56.0, 85.8)	71.8 (55.5, 90.7)	0.747	
**RANKL**	15.9 (10.7, 22.2)	15.9 (11.1, 24.0)	0.673	11

^†^ value/10^4^, ^‡^ Wilcoxon rank-sum test for comparison of the medians between two groups. *p*-values in bolds are statistically significant (*p* < 0.05) or close to being significant (*p* < 0.10).

**Table 3 biomedicines-11-02770-t003:** Distribution of the GCF biomarkers by periodontal status, N = 158.

Biomarker (pg/mL)	Low Tertile PD n = 52 (33%)	Medium/High Tertile PD n = 106 (67%)		
	Median (q1, q3)	Median (q1, q3)	*p*-Value ^§^	Missing Value
**IL-1a ^†^**	2.06 (0.76, 4.22)	4.36 (2.01, 8.36)	**0.0001**	
**IL-1b ^†^**	0.29 (0.10, 0.58)	0.59 (0.32, 1.33)	**0.0002**	
**IL-10 ^‡^**	0.34 (0.17, 0.80)	0.59 (0.29, 1.27)	**0.007**	1
**TNF-** **α**	0.31 (0.11, 0.67)	0.51 (0.17, 0.90)	**0.043**	3
**sICAM-1 ^†^**	1.95 (0.83, 3.75)	3.52 (1.44, 6.31)	**0.020**	
**sVCAM-1 ^†^**	0.27 (0.13, 0.48)	0.32 (0.11, 0.61)	0.560	12
**RANKL/OPG**	0.09 (0.05, 0.14)	0.09 (0.04, 0.16)	0.787	13

^†^ value/10^2^; ^‡^ value/SD; ^§^ Wilcoxon rank-sum test for comparison of the medians between two groups. *p*-values in bolds are statistically significant (*p* < 0.05) or close to being significant (*p* < 0.10).

**Table 4 biomedicines-11-02770-t004:** Odds ratios (OR), 95% CI of the associations between serum biomarkers and percent of sites with PPD ≥ 4 mm or percent of sites with CAL ≥ 4 mm.

	Low vs. Medium/High Percent of Sites with PPD ≥ 4 mm			
Biomarker (pg/mL)	Crude OR (95% CI)	*p*-Value	Adjusted OR (95% CI) ^‡^	*p*-Value
IL-1b	1.08 (0.97–1.21)	0.154	1.09 (0.97–1.24)	0.155
IL-6	1.03 (0.97–1.10)	0.357	1.04 (0.96–1.13)	0.364
IL-10	1.09 (1.01–1.17)	**0.028**	1.10 (1.01–1.20)	**0.033**
TNF-a	1.03 (0.98–1.09)	0.228	1.03 (0.97–1.09)	0.384
sICAM-1 ^†^	1.01 (1.00–1.01)	**0.074**	1.01 (1.00–1.02)	**0.060**
sVCAM-1 ^†^	1.00 (0.99–1.01)	0. 699	1.00 (0.99–1.01)	0.798
RANKL	1.00 (0.99–1.00)	0.440	1.00 (0.99–1.00)	0.376
	**Low vs. Medium/High Percent of Sites with CAL ≥ 4 mm**			
**Biomarker (pg/mL)**				
IL-1b	1.07 (0.95–1.19)	0.251	1.08 (0.95–1.23)	0.252
IL-6	1.00 (0.96–1.04)	0.994	0.99 (0.95–1.04)	0.668
IL-10	1.08 (1.00–1.17)	**0.038**	1.06 (0.98–1.16)	0.152
TNF-a	1.04 (0.98–1.10)	0.170	1.01 (0.95–1.08)	0.665
sICAM-1 ^†^	1.00 (1.00–1.01)	0.186	1.00 (1.00–1.01)	0.286
sVCAM-1 ^†^	1.00 (0.99–1.01)	0.672	1.00 (0.99–1.00)	0.338
RANKL	1.01 (0.99–1.03)	0.281	1.01 (0.99–1.04)	0.256

^†^ value/10^4^; ^‡^ adjusted for age, gender, education, smoking and alcohol status, BMI, mean plaque index, lipid-lowering agents (LLA) use, and HbA1c. *p*-values in bolds are statistically significant (*p* < 0.05) or close to being significant (*p* < 0.10).

**Table 5 biomedicines-11-02770-t005:** Odds ratios (OR), 95% CI of the associations between GCF biomarkers and percent of sites with PPD ≥ 4 mm or percent of sites with CAL ≥ 4 mm).

	Low vs. Medium/High Percent of Sites with PPD ≥ 4 mm			
Biomarker (pg/mL)	Crude OR (95% CI)	*p*-Value	Adjusted OR (95% CI) ^§^	*p*-Value
IL-1a ^†^	1.19 (1.07–1.32)	**0.001**	1.16 (1.04– 1.30)	**0.008**
IL-1b ^†^	2.91 (1.49–5.67)	**0.002**	2.40 (1.16–4.99)	**0.019**
IL-10 ^‡^	1.30 (0.85–2.01)	0.227	1.22 (0.76–1.97)	0.404
TNF-a	1.51(0.84–2.71)	0.169	1.48 (0.80–2.76)	0.213
sICAM-1 ^†^	1.00 (1.00–1.00)	0.067	1.00 (1.00–1.00)	0.102
sVCAM-1 ^†^	1.00 (1.00– 1.01)	0.337	1.00 (0.99–1.01)	0.601
RANKL/OPG	0.49 (0.13–1.89)	0.300	0.20 (0.04–1.10)	**0.064**
	**Low vs. medium/high percent of sites with CAL ≥ 4 mm**			
**Biomarker (pg/mL)**				
IL-1a ^†^	1.13 (1.04–1.23)	**0.005**	1.12 (1.02– 1.23)	**0.017**
IL-1b ^†^	2.37 (1.33–4.24)	**0.004**	2.09 (1.10–3.99)	**0.025**
IL-10 ^‡^	1.54 (0.95– 2.50)	0.078	1.48 (0.90–2.44)	0.120
TNF-a	1.43 (0.83–2.48)	0.199	1.43 (0.80–2.56)	0.228
sICAM-1 ^†^	1.00 (1.00–1.00)	0.977	1.00 (1.00–1.00)	0.761
sVCAM-1 ^†^	1.00 (0.99– 1.00)	0.408	0.99 (0.98–1.00)	**0.064**
RANKL/OPG	0.35 (0.09–1.41)	0.139	0.19 (0.04–0.94)	**0.041**

^†^ value/10^2^; ^‡^ value/SD; ^§^ adjusted for age, gender, education, smoking and alcohol status, BMI, mean plaque, index, lipid-lowering agents (LLA) use, and HbA1c. *p*-values in bolds are statistically significant (*p* < 0.05) or close to being significant (*p* < 0.10).

## Data Availability

The data and/or materials presented in this study are available upon request from the co-author, Dr. Oelisoa M Andriankaja (oelisoa.andriankaja@uky.edu). The data are not publicly available due to privacy and ethical restrictions.

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
