# Peer review of "Periodontal Disease, Local and Systemic Inflammation in Puerto Ricans with Type 2 Diabetes Mellitus"

_biomedicines, 2023, doi:10.3390/biomedicines11102770_

Round 1

Reviewer 1 Report

Comments for biomedicines-2603728

Authors demonstrated that serum and GCF endothelial and inflammatory mediators correlated with periodontitis determined with non-CDC/APA criteria. The research findings are compelling, and the overall study design is sound. However, the discussion section seems to be based on the assumption that periodontal disease elevates endothelial and inflammatory mediators in GCF or serum. This leads to some inconsistencies in the logical flow of the argument. The authors should focus on discussing the results that were actually obtained. To enhance readability, certain issues need to be addressed.

Study population

The participants in the study were also enrolled in the "Lipid-Lowering Agents Use in Periodontitis and Diabetes Study." While the authors argued in lines 190-196 that lipid-lowering agents should be considered as major non-dental factors for periodontal disease, their use was not included in the statistical analysis, as indicated in Tables 3 and 4. Recent narrative review suggested that lipid-lowering agents modulate inflammatory factors (Nutrients. 2023 Mar; 15(6): 1517). In addition, the article reported by the authors argued that lipid-lowering agents may reduce both systemic and oral inflammatory responses (J Clin Periodontol. 2015 42(12): 1090–1096.). The authors should present results adjusted for the use of lipid-lowering agents.

Table 4a and 4b. Some biomarkers were not included in the statistical analysis results because they are "already known to increase." However, the authors base their arguments on an analysis that compares the lowest tertile of the distribution to the medium or highest tertiles in one group. Therefore, these results should be included in the table. Additionally, table headers should “Low/medium vs. high percent of …”

The authors should discuss the strengths of their study using criteria other than CDC/APA. The CDC/APA criteria are clinical tools for the treatment of periodontal disease and are not intended for public health evaluation.

Author Response

Comments and Suggestions for Authors

Study population

1. a. The participants in the study were also enrolled in the "Lipid-Lowering Agents Use in Periodontitis and Diabetes Study." While the authors argued in lines 190-196 that lipid-lowering agents should be considered major non-dental factors for periodontal disease, their use was not included in the statistical analysis, as indicated in Tables 3 and 4. A recent narrative review suggested lipid-lowering agents modulate inflammatory factors (Nutrients. 2023 Mar; 15(6): 1517). In addition, the article reported by the authors argued that lipid-lowering agents may reduce both systemic and oral inflammatory responses (J Clin Periodontol. 2015 42(12): 1090–1096.). The authors should present results adjusted for the use of lipid-lowering agents.

==>Thank you for your recommendation. In line with your suggestion, we updated each estimate in Tables 3 and 4 by adding the lipid-lowering agents as a covariate in each model.

1. b. Table 4a and 4b. Some biomarkers were not included in the statistical analysis results because they are "already known to increase." However, the authors base their arguments on an analysis that compares the lowest tertile of the distribution to the medium or highest tertile in one group. Therefore, these results should be included in the table. Additionally, table headers should "Low/medium vs. high percent of …"

==>  Thank you. We have now made the requested revisions. Results from the analysis of the increase in the level of each biomarker (continuous form) or the elevated level of each biomarker determined by lowest/medium tertile vs. highest tertile of the biomarker's distribution (categorical form) are now displayed in Tables 4a and 4b.

1.c. The authors should discuss the strengths of their study using criteria other than CDC/APA. The CDC/APA criteria are clinical tools for the treatment of periodontal disease and are not intended for public health evaluation.

==> The data in this study were collected before the most recent classification of periodontal diseases. Therefore, we cannot change the periodontal parameters. However, in line with the reviewer's suggestion, a statement regarding the strengths of using periodontal parameters, such as percentage of sites with PPD≥4mm or percentage of sites with CAL≥4 mm such was added to the manuscript as follows:

" The data in this study were collected before the most recent and inclusive classification of periodontal and peri-implant diseases in 2017.[61] While the prevalence of periodontal disease can differ across the varieties of periodontitis and gingivitis case definitions, such as the CDC/AAP [62] or the 2017 AAP/EFP periodontitis classifications, the use of percentage of diseased sites based on the periodontal pocket depth and clinical attachment levels instead intend to express (or determine) the extent of existing periodontal inflammation, such as the focus of our study. In addition, these parameters are typically used for each individual treatment plan and for mechanistic studies.

Reviewer 2 Report

This manuscript entitled of “Periodontal disease, local and systemic inflammation in Hispanics with Type 2 diabetes mellitus aimed to assess the association between serum or gingival crevicular fluid endothelial and inflammatory mediators and chronic periodontal disease among type 2 diabetes mellitus Hispanic adults. Consequently, the authors suggested associations between oral and systemic endothelial and inflammatory mediators and periodontal clinical parameters among Hispanic adults with type 2 diabetes mellitus. The experiment performed relatively well, but there are some issues that need to be addressed.

1. In order to compare with other reported data and resolve inconsistencies, this study requires additional subcategories and analysis results according to drugs taken to treat diabetes.

2. Results and descriptions of IL-6, one of the key factors for identifying the TNF-alpha signaling pathway, are lacking.

3. There may be variables other than the race of the sample population, but there is a general lack of explanation for discrepancies between results compared to other reports.

Author Response

Comments and Suggestions for Authors

This manuscript titled  "Periodontal disease, local and systemic inflammation in Hispanics with Type 2 diabetes mellitus" aimed to assess the association between serum or gingival crevicular fluid endothelial and inflammatory mediators and chronic periodontal disease among type 2 diabetes mellitus Hispanic adults. 

Consequently, the authors suggested associations between oral and systemic endothelial and inflammatory mediators and periodontal clinical parameters among Hispanic adults with type 2 diabetes mellitus. The experiment performed relatively well, but there are some issues that need to be addressed.

1. In order to compare with other reported data and resolve inconsistencies, this study requires additional subcategories and analysis results according to drugs taken to treat diabetes.

==> Most of the variables on medications used to treat diabetes, such as metformin or insulin (injectable), and even variables related to the duration of diabetes are correlated with 'HbA1c'. We chose to use the variable 'HbA1c" as it is the most accurate and has complete data (i.e., no missing values). In addition, we have tried to add each variable one at a time in the models, and the variable 'Hba1c" contributed the most to the significant changes in the estimate of the associations compared to the other medications related to diabetes or other variables.

Additional subcategory analysis by medications used to treat diabetes reduces the sample size in each category, which further introduces a new bias as the study sample size is already small.

Moreover, most of those previously reported studies did not have detailed information on diabetes medication use or other medications, and inconsistencies remain between the studies.

Therefore, we included a brief paragraph related to this issue as a limitation of the study in the discussion section as follows:

We examined various variables related to diabetes treatment, including medications, such as metformin and injectable insulin, as well as diabetes duration. These variables were correlated with 'HbA1c' and explained less of the outcomes than HbA1c. Therefore, we utilized the 'HbA1c' variable due to having fewer missing values compared to the other covariates. Furthermore, when we considered each covariate for a potential adjustment in our statistical models, 'HbA1c' had the most significant impact on altering the associations when compared to other covariates. Due to statistical power constraints, subcategory analyses are discouraged as it might introduce a new bias, and hence reported as a limitation.”

2. Results and descriptions of IL-6, one of the key factors for identifying the TNF-alpha signaling pathway, are lacking.

==>  Serum level of IL-6 data was available and added to the study.

3. There may be variables other than the race of the sample population, but there is a general lack of explanation for discrepancies between results compared to other reports.

==>  Possible explanations for discrepancies between results across the studies were scatterly described throughout the discussion section, and we reorganized it by regrouping the information into a new paragraph as follows:

“Other than the race of the study sample, discrepancies between findings across the cited studies and ours might have been related to the differences in sample size, the study design, study populations, heterogeneity in periodontitis case definition(s) or periodontal parameter(s) employed, potential biases, and uncontrolled confounding factors. Most of the reports from the cited studies mentioned a small sample size, even if the report in question was based on the pre-PD-treatment baseline data of a clinical trial on individuals with T2DM, thus limiting the interpretation of their findings. The other reasons for negative results could be the presence of unknown uncontrolled confounding factors due to combined effects of multiple medications on the levels of the inflammatory mediators, which are difficult to disentangle.”

Round 2

Reviewer 1 Report

Authors addressed the issues which the reviewer commented. There are no additional comments for the manuscript.